# Universal Single-Mode Lasing in Fully Chaotic Billiard Lasers

**DOI:** 10.3390/e24111648

**Published:** 2022-11-14

**Authors:** Mengyu You, Daisuke Sakakibara, Kota Makino, Yonosuke Morishita, Kazutoshi Matsumura, Yuta Kawashima, Manao Yoshikawa, Mahiro Tonosaki, Kazutaka Kanno, Atsushi Uchida, Satoshi Sunada, Susumu Shinohara, Takahisa Harayama

**Affiliations:** 1Department of Applied Physics, School of Advanced Science and Engineering, Waseda University, 3-4-1 Okubo, Shinjuku-ku, Tokyo 169-8555, Japan; 2Department of Information and Computer Sciences, Saitama University, 255 Shimo-okubo, Sakura-ku, Saitama City 338-8570, Saitama, Japan; 3Faculty of Mechanical Engineering, Institute of Science and Engineering, Kanazawa University, Kanazawa 920-1192, Ishikawa, Japan; 4Department of Production Systems Engineering and Sciences, Komatstu University, Nu 1-3 Shicho-machi, Komatsu 923-8511, Ishikawa, Japan

**Keywords:** quantum chaos, billiard lasers, microcavity lasers

## Abstract

By numerical simulations and experiments of fully chaotic billiard lasers, we show that single-mode lasing states are stable, whereas multi-mode lasing states are unstable when the size of the billiard is much larger than the wavelength and the external pumping power is sufficiently large. On the other hand, for integrable billiard lasers, it is shown that multi-mode lasing states are stable, whereas single-mode lasing states are unstable. These phenomena arise from the combination of two different nonlinear effects of mode-interaction due to the active lasing medium and deformation of the billiard shape. Investigations of billiard lasers with various shapes revealed that single-mode lasing is a universal phenomenon for fully chaotic billiard lasers.

## 1. Introduction

Two-dimensional (2D) billiard lasers have been widely investigated over the past decades with various billiard shapes [1,2,3,4,5,6,7,8,9,10,11,12,13,14,15,16,17,18,19,20,21,22,23,24,25,26,27,28,29,30,31,32,33,34,35,36,37,38,39]. From the viewpoint of the property of ray dynamics which is equivalent to the motion of a point particle on a billiard table, 2D billiard lasers can be classified into three categories: integrable, partially chaotic, and fully chaotic billiard lasers [40]. On the basis of this classification of dynamical systems, the universality of energy level statistics was discovered by Casati et al. for the first time for a quantum billiard, which is a quantum particle on a billiard table where the motion of the corresponding classical point particle is fully chaotic [41]. This discovery led to the Bohigas–Gianonni–Schmidt (BGS) conjecture [42], and made significant contributions to the establishment of the research field of quantum chaos [43,44].

Although optical billiards can be viewed as a realization of quantum billiards, there is a noticeable difference between them. The light field inside an optical billiard is confined by the refractive index difference between inside and outside the optical billiard, and the corresponding rays obey Fresnel’s law at the billiard boundary. Meanwhile, quantum particles are completely confined inside the billiard table by the 2D rigid wall potential, and the corresponding classical point particles obey full reflection at the billiard boundary. As a result, eigenstates of a quantum billiard are bound states with real eigenenergies, whereas those of an optical billiard are quasibound states, or resonances, with complex eigenfrequencies because of non-Hermitian boundary conditions [1,2], where their real parts represent oscillation frequencies and their imaginary parts decay or loss rates of modes. In spite of the difference mentioned above, low-loss (i.e., well-confined) resonance wave functions in an optical billiard have properties similar to those of the eigenfunctions of the corresponding quantum billiard. Hence, these phenomena which were found for chaotic quantum billiards have been also observed in chaotic billiard lasers [3,4,7,10,11,12,17,20,21,26]. In this sense, chaotic billiard lasers have provided a valuable platform where the phenomena theoretically predicted in the research field of quantum chaos can be experimentally investigated.

When optical billiards are used as laser cavities (billiard lasers, in short), they exhibit essentially different properties from those of quantum billiards. That is, the resonance modes interact with each other through the active lasing medium in billiard lasers, which causes nonlinear dynamics of the light field [1,6,8,9,14,16]. In the spirit of the quantum chaos theory pioneered by Casati, it is important to turn our attention to universal properties of chaotic billiard lasers that resulted from the synergistic effect of the deformation of the billiard shape and nonlinear dynamics of resonance modes such as mode competition.

As an example of such a universal phenomenon for fully chaotic billiard lasers, it has been reported for the experiments of semiconductor billiard lasers with high pumping conditions that only single-mode lasing was observed for fully chaotic billiard lasers with stadium shapes with various aspect ratios, whereas only multi-mode lasing for integrable billiard lasers with elliptic shapes of different aspect ratios [28,32]. In addition, by assuming the similarity of wave functions for fully chaotic billiard lasers, it has been theoretically shown that single-mode lasing states are stable, whereas multi-mode lasing states are unstable when the billiard size is much larger than the wavelength and the external pumping power is sufficiently large [33]. Accordingly, in analogy with the universality that Casati found for quantum billiards [41], it can be expected that single-mode lasing is a universal phenomenon for fully chaotic billiard lasers which does not depend on the details of the billiard lasers. Although lasing characteristics of 2D billiard lasers with various 2D shapes have been intensively studied, universality has not been much pursued so far. Therefore, it is important to verify this universality by numerically studying fully chaotic billiard lasers with different shapes in the same way as the BGS conjecture was numerically examined for various systems [42].

In this paper, we numerically investigate the laser dynamics of three different shapes of fully chaotic billiard lasers; cardioid, D-shaped, and stadium billiard lasers shown in Figure 1, whose ray-dynamical trajectories have been exactly proven to be fully chaotic [45,46]. These billiards have already been studied in the research fields of quantum chaos and chaotic billiard lasers [8,9,13,15,17,19,28,30,32,37,41,42]. It is numerically shown that the single-mode lasing states are stable in all these three billiards in high pumping regimes, which verifies the universal single-mode lasing in fully chaotic billiard lasers [28,33]. We explain the detailed transition mechanism from multi to single-mode lasing, revealing the mode-pulling and the mode-pushing interaction among lasing modes. We also experimentally show that the numbers of lasing modes of the semiconductor cardioid, D-shaped, and stadium lasers are clearly different from those of semiconductor microdisk lasers, which also experimentally verifies the universality of single-mode lasing in fully chaotic billiard lasers.

The subsequent sections of this paper are organized as follows. In Section 2, we review the theoretical model to describe the dynamics of the light field and the lasing medium in billiard lasers and show numerical results of the resonances and wave functions for the cardioid billiard. In Section 3, the single-mode lasing is achieved in the course of time evolution of a numerically calculated light field in the cardioid billiard laser, whereas multi-mode lasing in the elliptic billiard laser when the pumping power is much larger than the lasing threshold. In Section 4, it is shown that the properties of the resonance wave functions of the cardioid and elliptic billiards are different from each other. We numerically obtain phase diagrams for the cardioid, D-shaped, and stadium billiard lasers, which verify the universal single-mode lasing. In Section 5, we provide experimental data for semiconductor billiard lasers with the cardioid, D-shaped, stadium, and circular shapes, which also validate the universal single mode lasing for fully chaotic billiard lasers. In Section 6, we numerically demonstrate that the occasional appearance of a couple of peaks experimentally observed in the spectra of the semiconductor fully-chaotic billiard lasers is attributed to the fluctuation of the temperature. Summary and discussion are given in Section 7.

## 2. The Schrödinger–Bloch
Model and the Resonances of the Cardioid Billiard Laser

We assume that the billiard laser is wide in the xy-directions and thin in the *z*-direction. As a result, the electromagnetic fields are separated into transverse magnetic (TM) and transverse electric (TE) modes. Here, we focus only on TM modes, whose electric field vector is expressed as *E*=$\left(0,0,E_{z}\right)$. The atoms in the lasing medium are assumed to have spherical symmetry and two energy levels. The relaxation due to the interaction with the reservoir can be described phenomenologically with decay constants $\gamma_{⊥}$ for the microscopic polarization ρ and $\gamma_{\|}$ for the population inversion *W*. The effect of the external energy injected into the lasing medium is introduced phenomenologically and represented by the pumping power $W_{\infty }$.

The slowly varying envelope approximation reduces the Maxwell equations as follows:(1)$$\frac{\partial \tilde{E}}{\partial t}=\frac{i}{2}\left(\nabla_{xy}^{2}+\frac{n^{2}}{n_{\mathrm{in}}^{2}}\right)\tilde{E}-\alpha_{L}\left(x,y\right)\tilde{E}+\frac{2\pi N\kappa \hbar }{\varepsilon }\tilde{\rho },$$
where $\tilde{E}$ and $\tilde{\rho }$ are, respectively, the slowly varying envelopes of the *z*-component of the electric field and the microscopic polarization, and *N* is the number density of the atoms, κ is the coupling strength, ε is the permittivity, and $\alpha_{L}$ represents the losses describing absorption inside the billiard and it equals to zero outside the billiard. The refractive index n(x,y) equals to $n_{\mathrm{in}}$ when the position (x,y) is inside the billiard and $n_{\mathrm{out}}$ outside the billiard. In the above, space and time are made dimensionless by the scale transformation $(\left(n_{in}\omega_{s}/c\right)x,$
$\left(n_{in}\omega_{s}/c)y\right)$→(x,y), $t\omega_{s}\to$*t*, respectively, where $\omega_{s}$ is the oscillation frequency of the fast oscillation part of the electric field.

The optical Bloch equations describing the active gain medium in interaction with the light field are written as
(2)$$\frac{\partial }{\partial t}\tilde{\rho }=-{\tilde{\gamma }}_{⊥}\tilde{\rho }-i\Delta_{0}\tilde{\rho }+\tilde{\kappa }W\tilde{E},$$
and
(3)$$\frac{\partial }{\partial t}W=-{\tilde{\gamma }}_{\|}(W-W_{\infty })-2\tilde{\kappa }(\tilde{E}{\tilde{\rho }}^{*}+{\tilde{E}}^{*}\tilde{\rho }),$$
where ${\tilde{\gamma }}_{⊥}\equiv \gamma_{⊥}/\omega_{s}$ and ${\tilde{\gamma }}_{\|}\equiv \gamma_{\|}/\omega_{s}$ respectively represent the dimensionless transversal and longitudinal relaxation rates, and $\Delta_{0}\equiv \left[\omega_{0}-\omega_{s}\right]/\omega_{s}$ represents the gain center, and $\omega_{0}$ is the transition frequency of the two-level atoms, and $\tilde{\kappa }\equiv \kappa /\omega_{s}$ represents the dimensionless couple strength. $W_{\infty }$ represents the external pumping power. The set of Equations (Equation 1)–(Equation 3) is called the Schrödinger–Bloch (SB) model [1,6,8,9,14]. The SB model can be numerically solved by the split-operator method (or so-called symplectic-integrator method). Therefore, it is more convenient for numerical simulations than the Maxwell–Bloch model.

By assuming the stationary oscillation of the electric and polarization field, respectively, $\tilde{E}={\tilde{E}}_{s}e^{-i\Delta_{s}t}$, $\tilde{\rho }={\tilde{\rho }}_{s}e^{-i\Delta_{s}t}$, and the constant population inversion, we obtain the time-independent SB equation,
(4)$$\left[\nabla_{xy}^{2}+\left(2\Delta_{s}+\frac{n^{2}}{n_{in}^{2}}\right)\right]{\tilde{E}}_{s}=2i\alpha \left(\Delta_{s}\right)\frac{W_{\infty }}{1+\eta \left(\Delta_{s}\right){\left|{\tilde{E}}_{s}\right|}^{2}}{\tilde{E}}_{s}-2i\alpha_{L}{\tilde{E}}_{s},$$
where the complex Lorentzian gain $\alpha (\Delta_{s})$ and the strength of the nonlinearity $\eta (\Delta_{s})$ are defined as follows:(5)$$\alpha \left(\Delta_{s}\right)\equiv \alpha_{0}\frac{1}{{\tilde{\gamma }}_{⊥}^{2}+{\left(\Delta_{s}-\Delta_{0}\right)}^{2}}\left({\tilde{\gamma }}_{⊥}+i\left(\Delta_{s}-\Delta_{0}\right)\right),$$
and
(6)$$\eta \left(\Delta_{s}\right)\equiv \frac{4{\tilde{\kappa }}^{2}}{{\tilde{\gamma }}_{//}{\tilde{\gamma }}_{⊥}}\frac{{\tilde{\gamma }}_{⊥}^{2}}{{\tilde{\gamma }}_{⊥}^{2}+{\left(\Delta_{0}-\Delta_{s}\right)}^{2}},$$
where $\alpha_{0}\equiv 2\pi N\kappa \tilde{\kappa }\hbar /\varepsilon$. The right-hand side of Equation (Equation 4) represents the effect of the lasing medium and vanishes outside the billiard.

The resonance wave function corresponding to the resonance (i.e., complex eigenfrequency) $\Delta_{j}$ is the solution of Equation (Equation 4) with $W_{\infty }=0$ and $\alpha_{L}=0$. When the pumping power $W_{\infty }$ is sufficiently increased from zero, the low-loss resonance closest to the gain center $\Delta_{0}$ changes into a stationary lasing state of the SB model. By linearizing the SB model equations, the lasing condition for the resonance $\Delta_{j}$ is obtained from the energy balance condition that the gain due to the pumping exceeds the losses [14]. Its mathematical expression is given by
(7)α0γ˜⊥W∞γ˜⊥2+(Δ0−ReΔj)2>−ImΔj+αL,
where the left-hand side represents the gain, and the right-hand side the losses. The frequency of the lasing mode is given by
(8)Δj′=γ˜⊥ReΔj+(−ImΔj+αL)Δ0−ImΔj+αL+γ˜⊥,
which is generally shifted from the frequency of the resonance ReΔj.

Figure 2 shows the distribution of the resonances of the cardioid optical billiard numerically obtained by the boundary element method [47] and the gain function for the SB model defined by gΔ=α0γ˜⊥/{γ˜⊥2+(Δ0−ReΔ)2}. The shape of a cardioid optical billiard is defined as r=R(1+cosθ) in the 2D polar coordinates with *R* = 30 in our numerical simulations as shown in Figure 1. Due to the symmetry of the cardioid billiard, the resonances are classified into two different classes of even and odd parities. The refractive index $n_{\mathrm{in}}$ is 3.3 inside the billiard and $n_{\mathrm{out}}\;=\;1$ outside the billiard. The other parameters are set as follows: $\Delta_{0}\;\;=\;\;0.04$, ${\tilde{\gamma }}_{⊥}\;=\;0.02$, $\alpha_{L}\;=\;0.004$, $\tilde{\kappa }\;=\;0.5$, ϵ=3.3, and Nκℏ=0.5. From Figure 2, one can see that there exist a number of resonances with low decay rates (i.e., |ImΔ|<0.01).

It has been theoretically shown that in a fully chaotic billiard laser, at least one single-mode lasing state is stable, whereas all multi-mode lasing states are unstable, when the external pumping power is sufficiently large and the billiard size is much larger than the wavelength so that ${\tilde{\gamma }}_{\|}\gg \left|\Delta_{ij}\right|$ holds, where $\Delta_{ij}$ is the difference between adjacent lasing frequencies [33]. This result provides a theoretical ground for recent experimental observations of universal single-mode lasing in fully chaotic billiard lasers [28,32]. We emphasize that the cardioid billiard laser numerically studied in this paper is much larger than those billiard lasers which have been studied so far. In addition, the gain band is much wider than those of previous studies. Therefore, the simulation condition shown in Figure 2 is more appropriate to numerically verify the universal single-mode lasing of fully chaotic billiard lasers. Figure 3 shows the spatial patterns of the resonance wave functions corresponding to the resonances labeled in Figure 2. All of these low-loss wave functions are distributed over the billiard domain and significantly overlap with one another reflecting fully chaotic ray-dynamical trajectories in the cardioid billiard [31,32,33,48].

## 3. Dynamical Simulation of Large Fully Chaotic and Integrable Billiard Lasers

Typical examples of the time evolution of the total light intensity inside the cardioid billiard laser are shown in Figure 4a,d,g. The longitudinal relaxation rate ${\tilde{\gamma }}_{\|}$ is set to 0.01, which is much larger than the frequency differences of adjacent resonances, i.e., ${\tilde{\gamma }}_{\|}\gg \left|\Delta_{ij}\right|$ (see also Figure 2). The lasing threshold $W_{\mathrm{th}}$ is found to be 0.000866. When the pumping power $W_{\infty }$ is 0.001 just above the threshold, the intensity converges to a certain constant after initial transient behavior as shown in Figure 4a. The power spectrum is obtained by the Fourier transformation of the time series in the stationary lasing regime in Figure 4a and has a single peak as shown in Figure 4b, which means that the stationary lasing state includes only one lasing mode. The spatial intensity pattern of this stationary single-mode lasing state is shown in Figure 4c. One can see that this intensity pattern is almost the same as that of the resonance wave function labeled by ➇ in Figure 3.

As the pumping power $W_{\infty }$ is increased, the other resonances become the stationary lasing modes which interact with one another due to the nonlinear effect of the lasing medium and the time evolution of the light intensity becomes oscillatory around a certain constant value as shown in Figure 4d where $W_{\infty }=0.005$. In this multi-mode lasing regime, the spectrum has several peaks as shown in Figure 4e and the final lasing state is not a stationary state but a limit-cycle or chaotic oscillation. Figure 4f shows a snapshot of the intensity pattern of the multi-mode lasing state.

When the pumping power $W_{\infty }$ is increased further, the intensities of the lasing modes increase much more, and hence, they strongly interact with one another, and finally, only one lasing mode wins the mode competition. The light intensity takes a constant value and the spectrum has a single peak as shown in Figure 4g,h as in the case of just above the threshold. However, typically, the stationary lasing state in this high pumping power regime is not a simple lasing mode corresponding to one resonance as in the case of just above the threshold but a locked state of a couple of resonances as shown in Figure 4i. One can see that the spatial pattern of this final stationary lasing state is asymmetric, violating the symmetry of the cardioid shape due to the spontaneous symmetry breaking caused by locking of resonance wave functions with different parities [9]. To summarize, the spectral peak in Figure 4b is attributed to a single resonance, but that in Figure 4h corresponds to a locked state of resonances.

The intensity pattern in Figure 4i is almost the same as that in Figure 5 which is obtained by superposing the resonance wave functions corresponding to the two resonances denoted by ➇ and ➈ in Figure 3. Accordingly, one can see that the final lasing state in the highly pumping regime is composed of two different resonance wave functions whose frequencies are originally different from each other. The frequency difference of these two modes vanishes because they are locked due to the mode pulling effect of the lasing medium. Consequently, the spectrum has a single peak and the light intensity takes a constant value instead of beating oscillatory behavior with two peaks in the spectrum. This is a typical transition process from multi-mode to single-mode lasing as the pumping power is increased in fully chaotic billiard lasers.

Next, we show laser dynamics of an elliptic billiard laser. The elliptic billiard shape for our numerical simulation is defined by $x^{2}+y^{2}/0.7^{2}=R_{0}^{2}$, where $R_{0}=43.9155$, so that this elliptic billiard has the same area as the cardioid billiard. Hence, the numbers of resonances are almost the same because of the Weyl formula. The longitudinal relaxation rate ${\tilde{\gamma }}_{\|}$ is set to 0.01, which is much larger than the frequency differences of adjacent resonances, i.e., ${\tilde{\gamma }}_{\|}\gg \left|\Delta_{ij}\right|$, that is, the same condition as that of the SB model simulation for the cardioid billiard lasers.

The wave functions corresponding to the bouncing ball trajectories have much larger decay rates than the whispering gallery modes which become stationary lasing states. Therefore, the resonance wave functions of an elliptic billiard which comprise the stationary lasing states are regularly localized inside the billiard according to the quantum numbers in the radial and rotational directions, which systematically determine the high-intensity positions in the spatial patterns of the resonance wave functions. Consequently, many lasing modes coexist without interacting with one another by avoiding the overlap of high-intensity regions of the wave functions, and hence, many peaks are observed in the spectrum as shown in Figure 6e. For the elliptic billiard laser, single-mode lasing is only observed just above the threshold $W_{\mathrm{th}}=0.000666$. Figure 6a–c show data for such single-mode lasing at $W_{\infty }=0.001$.

When the pumping power is increased further, those resonance wave functions which do not overlap with the dominant lasing modes become new stationary lasing modes and the number of the stationary lasing modes increases as shown in Figure 6g–i. Consequently, multi-mode lasing is always observed in the high pumping regime in integrable billiard lasers.

## 4. Spatial Overlap between Two Modes and Phase Diagram

Long-lived eigenstates in open chaotic mapping systems have been shown to be localized on the forward trapped set of the corresponding classical dynamics in the short-wavelength limit [49,50,51,52,53]. This localization is a manifestation of quantum ergodicity in open quantized classically fully-chaotic systems. Similar phenomena have also been observed in fully chaotic 2D optical billiards numerically [32], where resonance wave functions of low-loss modes are localized around the forward trapped set of the corresponding ray-dynamics with Fresnel’s law in the short wavelength limit as shown in Figure 7 [31,48,54]. In addition, assuming that all of the spatial patterns of the wave functions for low-loss modes in a fully chaotic billiard lasers are similar to one another, one can analytically prove the stability of a single-mode lasing state and instability of a multi-mode lasing state [33]. The overlap is an important quantity to measure the wave function similarity. It is defined by the following cross-correlation of the amplitude distributions between an arbitrary pair of resonance wave functions,
(9)$$C=\frac{\int |\phi (r)||\psi (r)|)dr}{\sqrt{\left({\int |\phi \left(r\right)|}^{2}dr)({\int |\psi \left(r\right)|}^{2}dr\right)}},$$
where ϕ and ψ are the resonance wave functions and the integrals are taken over the 2D billiard [32].

Figure 8 shows the histogram of the overlap *C* for the cardioid billiard, where most of the overlap values are found between 0.6 and 0.8. The functional form of the histogram is found to be universal for fully chaotic billiards [32,48]. For the elliptic billiard, the overlap *C* can take a smaller value and is distributed over a wider range. These results indicate that almost all resonance wave functions of the cardioid billiard lasers are similarly distributed inside the billiard, whereas the resonance wave functions of the elliptic billiard lasers exhibit various spatial patterns inside the billiard [31,33,48]. On the basis of quantum-classical correspondence in the semiclassical regimes, it is expected that the shorter the wavelength, the more distinct the histograms between fully chaotic and integrable billiards. This means that the effects caused by the spatial distributional difference of the resonances between fully chaotic and integrable billiards are expected to be more prominent for larger billiard lasers.

In addition, as the size of the billiard laser becomes larger, the frequency difference $\Delta_{ij}$ between adjacent resonances is decreased and becomes much smaller than the longitudinal relaxation rate ${\tilde{\gamma }}_{\|}$. Therefore, large billiards satisfy the condition ${\tilde{\gamma }}_{\|}\gg \left|\Delta ij\right|$. In order to verify the importance of this condition, we change the value of the longitudinal relaxation rate ${\tilde{\gamma }}_{\|}$ with fixing the size and pumping power of the cardioid billiard laser. As we have already shown in Figure 4g–i, single-mode lasing is observed with ${\tilde{\gamma }}_{∥}=0.01$ and $W_{\infty }\;=\;0.01$. Here, we decrease the value of ${\tilde{\gamma }}_{\|}$ without changing the remaining parameter values. When its value is decreased to 0.009, the other two peaks appear in the spectrum as shown in Figure 9a, where the left and right peaks respectively correspond to the locked state of the resonances ➃ and ➄ and the locked state of the resonances ⑬ and ⑭ (resonances are labeled in Figure 3). As the value of ${\tilde{\gamma }}_{\|}$ is decreased gradually to 0.007, more and more peaks appear as shown in Figure 9b,c. All of these peaks can also be assigned to the resonances in Figure 3 and some of them are the locked states of adjacent different parity resonance pairs. We confirmed that even with these smaller values of ${\tilde{\gamma }}_{\|}$, those peaks in the spectrum disappear and change into a single peak when the pumping power $W_{\infty }$ is increased. This way, we obtain the phase diagram of the lasing state in the ${\tilde{\gamma }}_{\|}$-$W_{\infty }$ plane shown in Figure 10a for the cardioid billiard laser. One can see that only single-mode lasing is always observed as far as ${\tilde{\gamma }}_{\|}$ and $W_{\infty }$ are sufficiently large.

Our simulations of the cardioid billiard laser revealed the existence of two types of single-mode lasing regimes in the ${\tilde{\gamma }}_{\|}$-$W_{\infty }$ plane. One regime appears just above the lasing threshold, where the lasing mode corresponds to a single resonance with the largest net gain. Another regime appears for large $W_{\infty }$ values, where the lasing mode corresponds to a locked mode of different symmetry-class resonances. We confirmed that these two types of single-mode lasing regimes were observed in the SB model simulations of the D-shaped billiard (R=30 and d=15) and the stadium billiard (R=30), as shown in Figure 10b,c. Figure 10 is our main result of this paper, showing that single-mode lasing of a locked mode is commonly observed for the three different fully chaotic billiard lasers when the pumping power is sufficiently large.

We note that although the SB model simulations reported in this paper are performed for larger sizes of billiards than those previously numerically studied, they are still not so large as the typical semiconductor lasers used in experiments. The sizes of billiards for the SB model simulation in this paper are estimated less than 10 μm if the wavelength is assumed 1 μm, which is typical for conventional semiconductor lasers while their typical sizes are much larger than 10 μm.

Therefore, in relatively large billiard lasers with linear dimensions more than 100 μm, the frequency differences $\Delta_{ij}$ between adjacent resonances are significantly smaller than those in the simulation of this paper, which implies that the transition threshold from multi-mode to single-mode lasing is much smaller than those in Figure 10. Consequently, it is expected that the universal single-mode lasing can be observed in the experiments of semiconductor fully-chaotic billiard lasers with typical sizes.

## 5. Experiments of Semiconductor Billiard Lasers

By applying a reactive-ion-etching technique to a graded index separate-confinement-heterostructure single quantum well GaAs/Al${}_{x}$Ga${}_{1-x}$As structure, we fabricated semiconductor cardioid, D-shaped, and stadium billiard lasers of which microscope images are shown in Figure 11a–c. We also fabricated the circular billiard laser, that is, the microdisk laser as the representative example of integrable billiard lasers shown in Figure 11d.

The lasers were soldered on aluminum nitride submounts, and electrically driven with current injection under continuous-wave (cw) operation. The temperature was kept at room temperature. The difference between cw and pulse operations is crucial for observing the universal single-mode lasing, because relaxation from initial transient multi-mode lasing to single-mode lasing needs a constant pumping with a certain duration. Thus, for pulse operations with pulse widths smaller than the above duration, one cannot observe the universal single-mode lasing [13,28,55].

To measure the optical spectra of the semiconductor billiard lasers, the emitted light was collected with anti-reflection-coated lenses and coupled to a multimode optical fiber via a 30 dB optical isolator. The multimode optical fiber was coupled to a spectrum analyzer (Advantest Q8347). Figure 12 shows the measured spectra of the semiconductor billiard lasers. Every spectrum of three different fully chaotic billiard lasers has a single peak with the injection currents much larger than their thresholds as shown in Figure 12a–c, whereas the spectrum of the integrable billiard (microdisk) laser has more than 10 peaks as shown in Figure 12d. The appearance of a single lasing peak in the spectra of the semiconductor fully-chaotic billiard lasers can be more convincingly confirmed with the high-resolution mode of the spectrum analyzer with wavelength interval of 0.002 nm. As shown in Figure 13, a narrow linewidth peak was measured in the semiconductor D-shaped billiard lasers.

To gain further insight in the spectral characteristics, we counted the number of the peaks whose intensities were larger than −35 dB of the maximum peak intensity in each spectrum. Figure 14 shows the injection–current dependence of the numbers of peaks in the spectra. From Figure 14, one can see that the number of the lasing modes in a fully chaotic billiard laser is distinctly different from that of the integrable one when the injection currents are much larger than their thresholds. Similar experimental results were reported for the semiconductor stadium and elliptic billiard lasers with various aspect ratios [32]. Accordingly, we conclude that this phenomenon is universal and independent of the details of billiard lasers. Thus, the universal single-mode lasing in fully chaotic billiard lasers is verified experimentally.

The spectra observed in the experiments of fully chaotic billiard lasers occasionally contain a couple of peaks instead of one single peak even when the injection current is much larger than the threshold as one can see in Figure 14. In the next section, we show by the SB model simulation that this phenomenon can be explained by the temperature fluctuation of the billiard lasers.

In the experiments, we always try to keep the temperature of the billiard lasers to a fixed value by the Peltier cooler with the thermostat. However, even if we set the temperature of the temperature control device to a fixed value, it is impossible to make the actual temperature of the laser strictly kept to be this value, and the fluctuation of temperature is practically unavoidable during the experiments. Temperature fluctuations affect the states of the lasing medium and change the gain profile. Since the sizes of the semiconductor fully-chaotic billiard lasers are large and the gain band contains a number of resonances, a slight change of the gain profile can change the stationary lasing states significantly.

In order to elucidate the temperature dependence of the stationary lasing states of the semiconductor stadium billiard laser, we investigate the change of spectral peaks by increasing the temperature from $20.1{\;}^{\circ }C$ to $21.0{\;}^{\circ }C$ by $0.1{\;}^{\circ }C$. The result is shown in Figure 15, where we find that the gradual temperature increase from $20.4{\;}^{\circ }C$ to $20.8{\;}^{\circ }C$ results in a drastic change of the lasing state (i.e., the appearance of a peak at 862 nm in addition to a peak at around 857 nm), whereas the other temperature changes do not cause significant spectral changes. The temperature increase shifts the gain center to the right, i.e., longer wavelengths, which explains the appearance of the peak at around 862 nm. What is interesting here is that only this peak is observed, despite there are a number of resonances between the two peaks, and the gain center is gradually moved from left to right. In the following section, we show that this experimental result can be reproduced by the SB model simulations. Namely, we numerically demonstrate that low-loss resonances very close to the gain center do not necessarily contribute to the formation of a final lasing state obtained for a high pumping power, and that there is a selected set of low-loss resonances that can contribute to the final lasing state.

## 6. Effect of Temperature Fluctuation

We investigate the dynamics of the D-shaped billiard laser by changing the gain center $\Delta_{0}$ in the SB model simulation with the fixed pumping power $W_{\infty }$. The size and deformation parameters of this billiard are R=30 and d=15, respectively, and the distribution of the resonances in the complex frequency plane is shown in Figure 16. When the gain centers are 0.181 and 0.22, the power spectra have single peaks, respectively corresponding to the locked state of the resonances A and B and the locked state of the resonances C and D as shown in Figure 17a,c. When the gain center is 0.206, the two locked states simultaneously appear in the power spectrum as shown in Figure 17b. This is explained by the fact that the net gains of two locked states are larger than those of the resonances around 0.206. Consequently, two distinct peaks are simultaneously observed in the spectra, which correspond to the experimental results shown in Figure 15.

Finally, the pumping power $W_{\infty }$ as well as the gain center $\Delta_{0}$ are increased. Even if the position of the gain center is located between the real parts of the resonances B and C, one of the two locked states corresponding to the two peaks in Figure 17b wins the mode competition and becomes a single-mode lasing state with large pumping power. We define this transition threshold $W_{t}$ as the value of the pumping power where the lasing state changes from the multi-mode to the single-mode lasing state. The gain center dependence of $W_{t}$ is shown in Figure 18. $W_{t}$ is small when the gain center is near the low-loss resonances such as 0.181 and 0.22. However, it becomes extremely large when the gain center is the midpoint between these two values, and we could not obtain the single-mode lasing state by the SB model simulation because the net gains of the locked state of A and B and the locked state of C and D are equally large. If the size of the billiard is much larger, the resonance distribution is much more dense. Therefore, the two-mode lasing regime in the pumping power is expected to be small in large fully-chaotic billiard lasers. However, when the injection current is large, the temperature of the upper contact layer might be very high due to the resistance there, and hence, the lower Peltier device might operate to cool the laser from the bottom and the whole situation might become a highly non-equilibrium state, which could result in the fluctuation of the gain center. Therefore, it might be inevitable to observe a couple of peaks occasionally in the spectra of the semiconductor fully-chaotic billiard lasers, which is the effect of the combination of the universal single-mode lasing and the fluctuation of the gain center.

## 7. Summary and Discussion

By the SB model simulation of the cardioid, D-shaped, and stadium billiard lasers, we verified the universal single mode lasing in fully chaotic billiard lasers: at least one single-mode lasing state is stable, whereas all multi-mode lasing states are unstable, when the external pumping power is sufficiently large and the billiard size is much larger than the wavelength so that ${\tilde{\gamma }}_{\|}\gg \left|\Delta_{ij}\right|$ is satisfied, where $\Delta_{ij}$ is the difference between adjacent lasing frequencies. The stability of multi-mode lasing states in integrable billiard lasers is also demonstrated by the SB model simulation of the elliptic billiard laser.

We also verified the universal single-mode lasing by the experiments of the semiconductor cardioid, D-shaped, stadium, and circular billiard lasers. The occasional observation of a couple of peaks instead of a single peak in the spectra of the semiconductor fully-chaotic billiard lasers was attributed to the fluctuation of the temperature on the basis of the SB model simulation. We expect that the occasional appearance of a couple of peaks in the spectra would vanish and only a single peak would be observed if the semiconductor fully-chaotic billiard lasers have much lower resistances and the control of the temperature is more accurate.

## Figures and Tables

**Figure 1 entropy-24-01648-f001:**
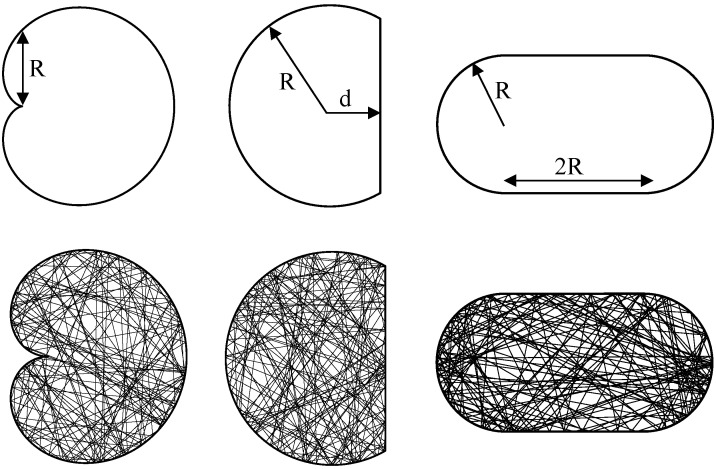
Fully chaotic billiards. From left to right: cardioid, D-shaped, and stadium.

**Figure 2 entropy-24-01648-f002:**
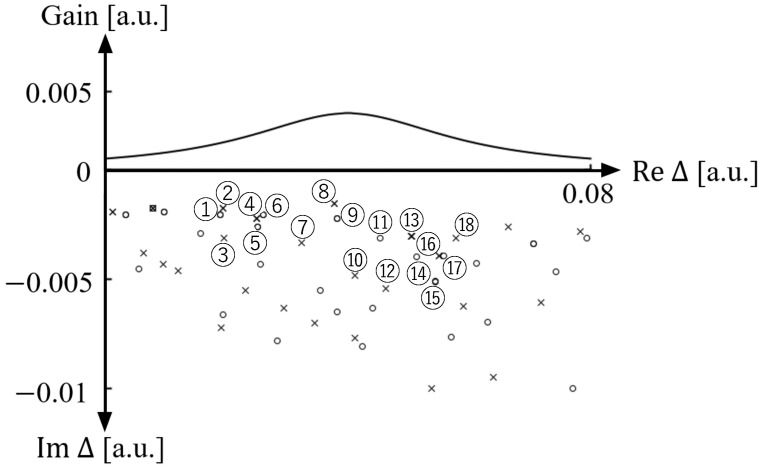
Gain and resonances in a cardioid billiard. Circles (∘) denote the resonances with odd parity while crosses (×) those with even parity.

**Figure 3 entropy-24-01648-f003:**
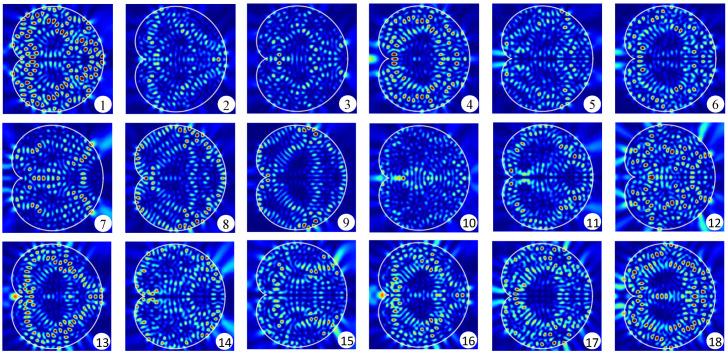
Spatial intensity patterns of eighteen low-loss resonances labeled in Figure 2.

**Figure 4 entropy-24-01648-f004:**
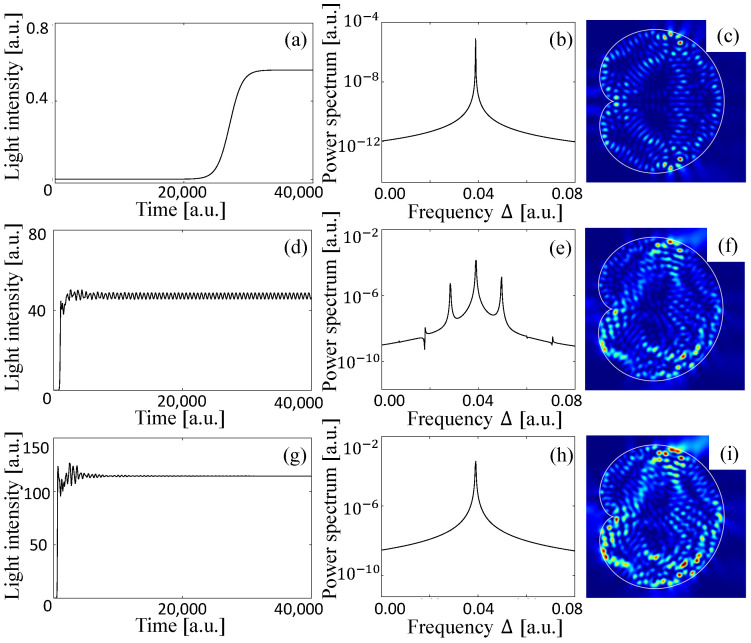
Dynamics of a cardioid billiard laser numerically simulated by the SB model. Time evolution of the total intensity inside the billiard with the pumping power $W_{\infty }\;=\;$ (**a**) 0.001, (**d**) 0.005, and (**g**) 0.010. (**b**,**e**,**h**) The spectrum obtained from the stationary oscillation regime with the corresponding pumping powers. (**c**,**f**,**i**) The intensity patterns of the final lasing states with the corresponding pumping powers.

**Figure 5 entropy-24-01648-f005:**
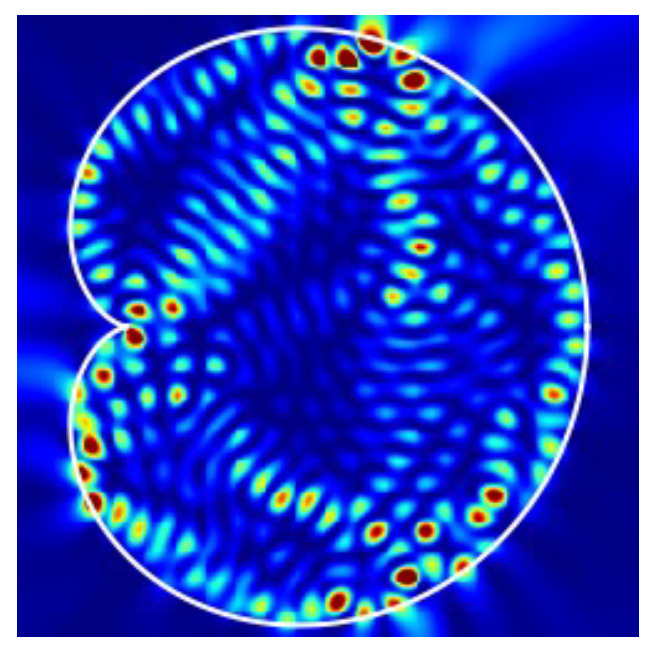
The spatial intensity pattern obtained by the superposition of the two resonances denoted by ➇ and ➈ in Figure 3, which have even and odd parity, respectively. The superposition of different-parity wave functions yields an asymmetric intensity pattern (see Ref. [9] for a detailed explanation).

**Figure 6 entropy-24-01648-f006:**
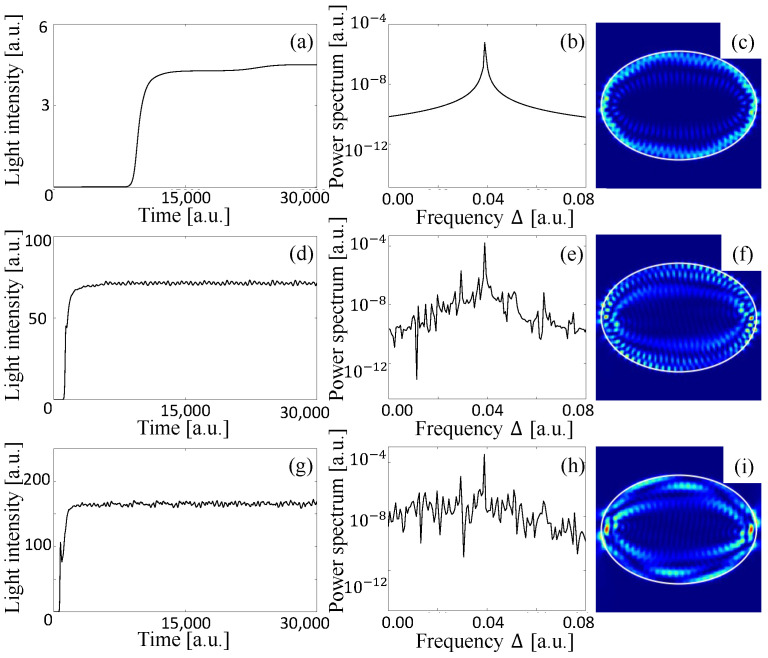
Dynamics of an elliptic billiard laser numerically simulated by the SB model. Time evolution of the total intensity inside the billiard with the pumping power $W_{\infty }\;=\;$ (**a**) 0.001, (**d**) 0.005, and (**g**) 0.010. (**b**,**e**,**h**) The spectrum obtained from the stationary oscillation regime with the corresponding pumping powers. (**c**,**f**,**i**) The intensity patterns of the final lasing states with the corresponding pumping powers.

**Figure 7 entropy-24-01648-f007:**
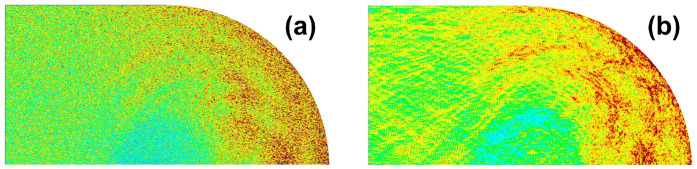
Numerically computed light intensity distributions inside the stadium optical billiard with refractive index 3.3 (because of the $C_{2v}$ symmetry of the stadium shape, only a quarter domain is shown). (**a**) Conditionally invariant measure obtained by ray simulation incorporating Fresnel’s law with 25 million initial conditions (See Ref. [48] for a detailed explanation). (**b**) Average of 30 low-loss resonance wave functions.

**Figure 8 entropy-24-01648-f008:**
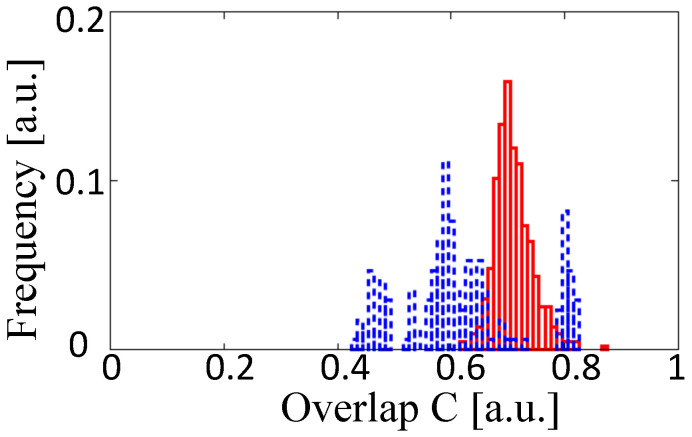
The histogram of the overlap *C* for the cardioid billiard (red solid line) and the elliptic billiard (blue dotted line) calculated by using 30 and 19 low-loss resonance wave functions, respectively.

**Figure 9 entropy-24-01648-f009:**
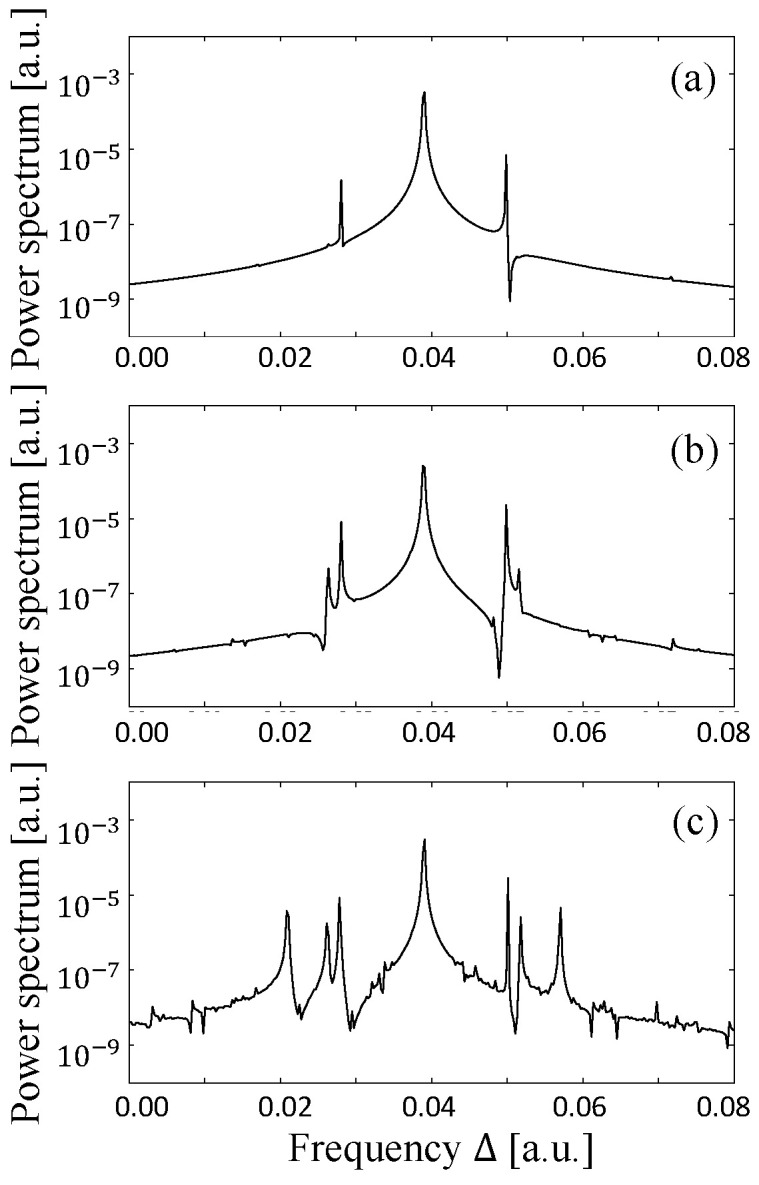
Power spectra obtained from the lasing states for the cardioid billiard laser with the fixed pumping power $W_{\infty }=9.1\;W_{th}$ and the values of the longitudinal relaxation rate $\tilde{\gamma }=$ (**a**) 0.009 (**b**) 0.008 (**c**) 0.007.

**Figure 10 entropy-24-01648-f010:**
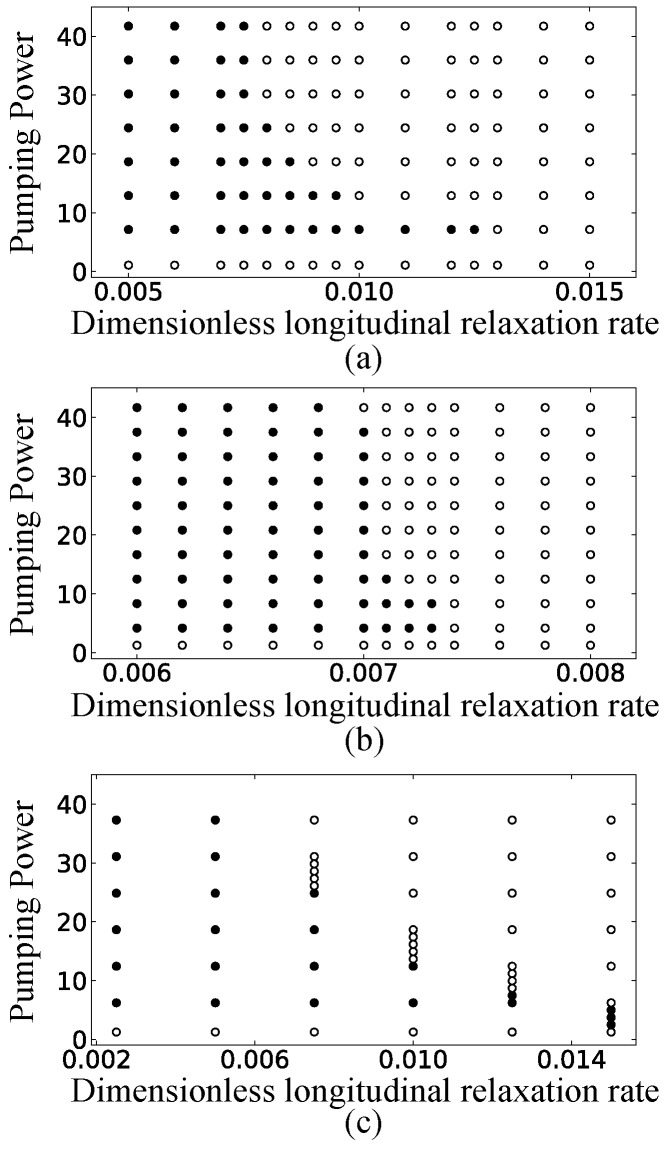
Phase diagrams of lasing states with various longitudinal relaxation rates ${\tilde{\gamma }}_{∥}$ and pumping powers $W_{\infty }/W_{\mathrm{th}}$ for (**a**) the cardioid billiard laser, (**b**) the D-shaped billiard laser, and (**c**) the stadium billiard laser. The white and black circles correspond to single-mode and multi-mode lasing states, respectively.

**Figure 11 entropy-24-01648-f011:**
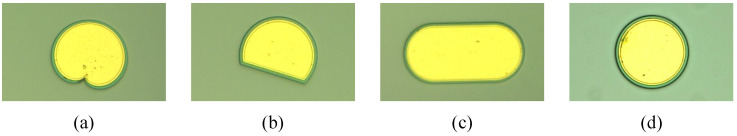
Optical microscope images of the fabricated lasers of (**a**) the cardioid billiard (*R* = 35 μm), (**b**) the D-shaped billiard (*R* = 40 μm, *d* = 20 μm), (**c**) the stadium billiard (*R* = 50 μm), and (**d**) the circular billiard, i.e., microdisk (*R* = 40 μm).

**Figure 12 entropy-24-01648-f012:**
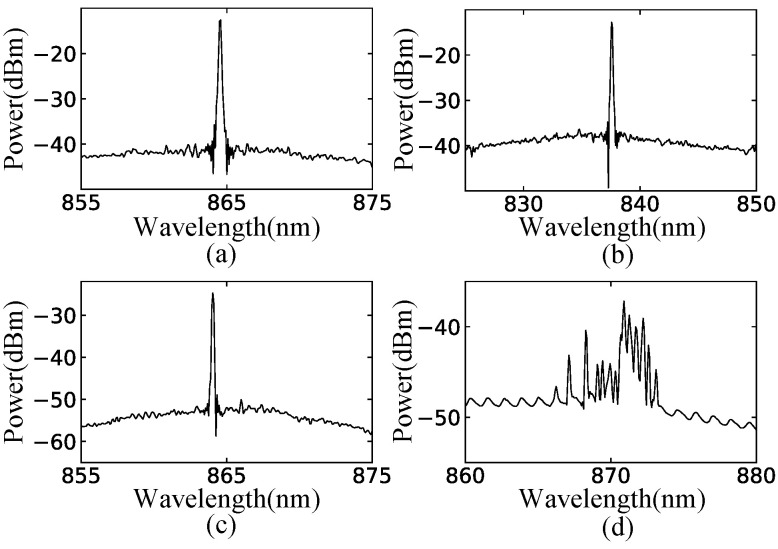
Spectra observed in the experiments of (**a**) the cardioid laser with the injection current 160 mA, (**b**) the D-shaped laser with the injection current 200 mA, (**c**) the stadium laser with the injection current 200 mA, and (**d**) the microdisk laser with the injection current 120 mA. The thresholds of these lasers are around 30 mA, 45 mA, 40 mA, and 20 mA, respectively.

**Figure 13 entropy-24-01648-f013:**
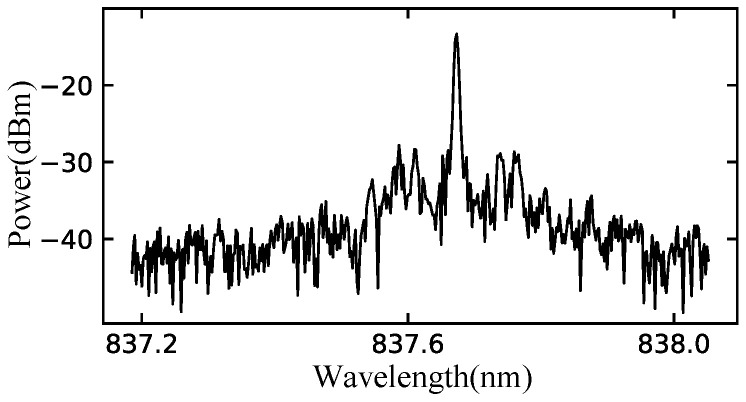
A spectrum obtained by using the spectrum analyzer in high-resolution mode with wavelength interval of 0.002 nm in the experiment of the semiconductor D-shaped billiard laser with the injection current 200 mA.

**Figure 14 entropy-24-01648-f014:**
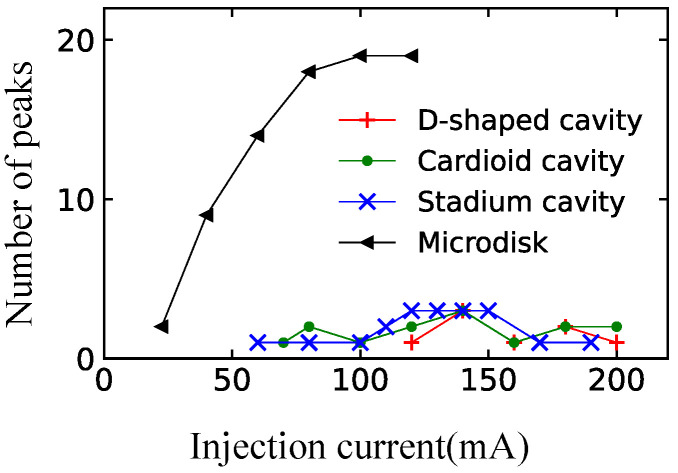
Injection current dependence of the number of the spectral peaks of fully chaotic and integrable billiard lasers.

**Figure 15 entropy-24-01648-f015:**
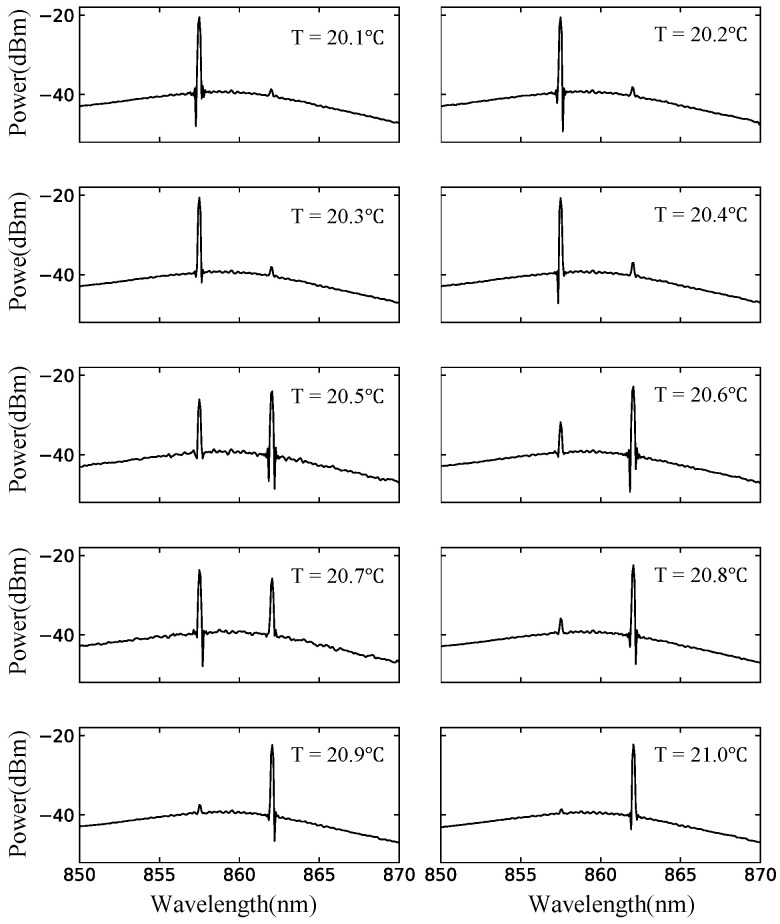
Spectra of the semiconductor stadium billiard laser with the injection current 80 mA at various temperatures between $20.1{\;}^{\circ }C$ and $21.0{\;}^{\circ }C$.

**Figure 16 entropy-24-01648-f016:**
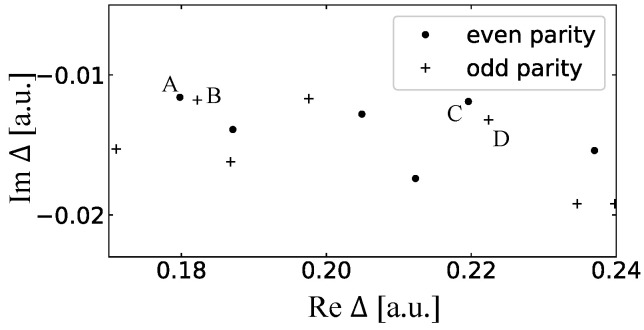
The resonance distribution for the D-shaped billiard.

**Figure 17 entropy-24-01648-f017:**
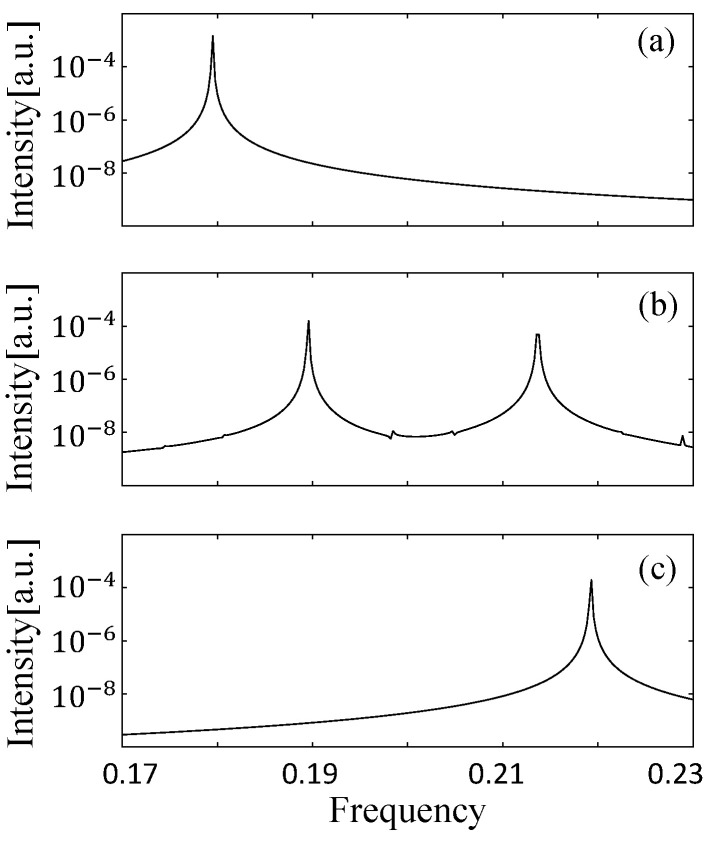
Power spectra of the lasing states with different gain centers $\Delta_{0}$ ($\tilde{\gamma_{∥}}\;=\;0.008$). (**a**) $\Delta_{0}\;=\;0.181$. (**b**) $\Delta_{0}\;=\;0.206$. (**c**) $\Delta_{0}\;=\;0.220$.

**Figure 18 entropy-24-01648-f018:**
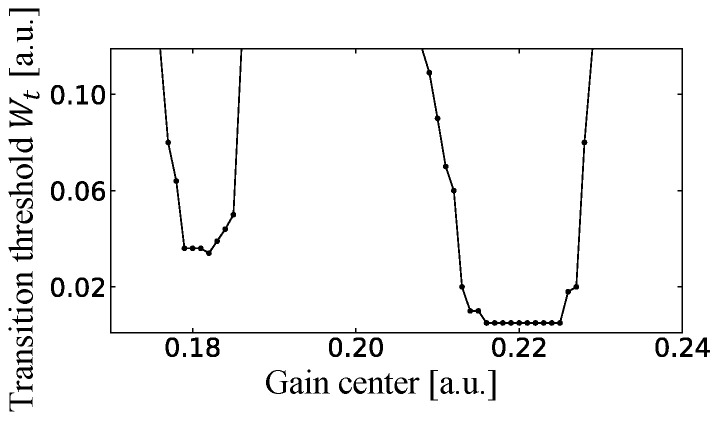
The gain center dependence of the transition threshold $W_{t}$ from multi-mode to single-mode lasing.

## Data Availability

Not applicable.

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
