# Peer review of "Universal Single-Mode Lasing in Fully Chaotic Billiard Lasers"

_entropy, 2022, doi:10.3390/e24111648_

Round 1

Reviewer 1 Report

Report for the Manuscript
Universal single-mode lasing in fully-chaotic billiard lasers
by Mengyu You et al.

The authors of the manuscript "Universal single-mode lasing in fully-chaotic billiard lasers" provide a very nice, comprehensive and detailed overview of lasing chaotic optical microcavities. It is ideally suited for this Special Issue. Not only is it at heart of quantum chaos, it also merges theory, numerics, and experiments in a single work.

The paper starts with a wonderful introduction into optical billiards, those lasing in particular, as one model systems for quantum chaos. The author introduce then the laser physics, leading to an interaction of modes (via the active medium) that is not present in hard-wall quantum billiards. So there are two nonlinear effects present in chaotic billiards lasers, one from their geometric shape and another one from the mode interaction. This interplay and the mutual effects between (chaotic or integrable) geometry and mode interaction are the subject of the manuscript under review, with the focus on finding out the universal effects - very much in the spirit of Casati.

The main result of the authors is a confirmation of universality, namely that single-mode lasing is an universal property of fully-chaotic billiard lasers confirmed by the detailed study of cardioid, D-shape, and stadium billiard lasers in comparison to elliptic lasers. The lasing properties are captured by the Schrödinger-Bloch model that is introduced in Section 2.

The following chapters explain in detail how the authors arrive at their conclusions, and show nice examples for all phenomena described, in particular the multimode vs. single mode lasing regimes in various situations. The paper ends with a compact summary.

There is one question from my side for the authors. It concerns Fig. 7(a), namely, how this ray simulation result was obtained, e.g. how the initial conditions were chosen. It a single ray shown or an ensemble? Is it possible to explain the somewhat curly structures in the ray model? (Also, would one naively not expect something more chaotic?) That would give a nice asset with respect to the wave simulations, beyond the very nice agreement that is clearly visible.

Without any hesitation I recommend publication of this nice work.

Author Response

We deeply thank the reviewer1 for her/his very positive review of our manuscript. Concerning her/his wonderful question, please see the attachment.

Reviewer 2 Report

Comment on 

Universal single-mode lasing in fully-chaotic billiard lasers 

by Mengyu You et al

The paper discussed universality of single-mode lasing in fully-chaotic optical microcavity billiards in the large size and the large pumping power limit. The proposal is based mainly on numerical experiments, but the author verified it in laboratory experiments with semiconductor billiard lasers.  The paper is clearly written and the results are interesting enough for publication.  Readers might be interested in the following issues: 

- In Fig.5, why is an asymmetric spatial pattern obtained in the superposition of the two resonances 8 and 9 in Fig.3, although both look symmetric with respect to the horizontal axis?

- From the difference of the resonance distribution patterns, it would be qualitatively reasonable to expect single-mode lasing in fully-chaotic situations. Is it possible to more sharply formulate the condition for single-mode lasing, say by distinguishing in terms of resonance distribution patterns? 

- For the mixed system in which the associated phase space is a mixture of regular and chaotic components, what happens? Is single-mode lasing  not observed, or does it depend on the situation? Alternatively, is full-chaoticity not only sufficient but necessary for single-mode lasing?

Author Response

We are really grateful to Reviewer 2 for her/his very positive review of our manuscript. Please see the attachment as for her/his thoughtful questions.
